# Chemically Insignificant Social Parasites Exhibit More Anti-Dehydration Behaviors than Their Hosts

**DOI:** 10.3390/insects12111006

**Published:** 2021-11-08

**Authors:** Maria Cristina Lorenzi

**Affiliations:** Laboratoire d’Ethologie Expérimentale et Comparée, LEEC, UR 4443, Université Sorbonne Paris Nord, F-93430 Villetaneuse, France; lorenzi@univ-paris13.fr

**Keywords:** water balance, cuticular hydrocarbons, paper wasps, *Polistes atrimandibularis*, *Polistes biglumis*, heat stress

## Abstract

**Simple Summary:**

Social parasites use a variety of deceptive mechanisms to avoid detection by their social-insect hosts and get tolerance in their colonies. One of these mechanisms is chemical insignificance, where social parasites have reduced amounts of recognition cues—hydrocarbons—on their cuticle, thus evading host chemical detection. This exposes social parasites to dehydration stress, as cuticular hydrocarbons also limit body water loss. By analyzing behavioral data from field observations, here we show that a *Polistes* wasp social parasite exhibits water-saving behaviors; parasites were less active than their cohabiting host foundresses, spent more time at the nest, and rested in the shadow, contradicting the rule that dominant individuals occupy prominent positions at the nest.

**Abstract:**

Social parasites have evolved adaptations to overcome host resistance as they infiltrate host colonies and establish there. Among the chemical adaptations, a few species are chemically “insignificant”; they are poor in recognition cues (cuticular hydrocarbons) and evade host detection. As cuticular hydrocarbons also serve a waterproofing function, chemical insignificance is beneficial as it protects parasites from being detected but is potentially harmful because it exposes parasites to desiccation stress. Here I tested whether the social parasites *Polistes atrimandibularis* employ behavioral water-saving strategies when they live at *Polistes biglumis* colonies. Observations in the field showed that parasites were less active than their cohabiting host foundresses, spent more time at the nest, and rested in the shadowy, back face of the nest, rather than at the front face, which contradicted expectations for the use of space for dominant females—typically, dominants rest at the nest front-face. These data suggest that behavioral adaptations might promote resistance to desiccation stress in chemical insignificant social parasites.

## 1. Introduction

Many organisms are attracted by the abundance and variety of resources accumulated in the nests of social insects—from food to large brood, from relatively well-controlled microclimatic conditions to protection from predators [1,2]. In addition, social insects have a specialized workforce, i.e., non-reproductive individuals which carry out nest maintenance and take care of allo-parental brood; the latter service is the target of permanent social parasites, which rely on their hosts for rearing their own brood [2,3,4].

Social parasite females (queens) invade and integrate into the host colony. This requires special adaptations because, as a result of a coevolutionary arms race against parasites, hosts have evolved effective defenses and reject most non-nestmates from their colonies [5,6,7,8].

As a front-line defense, hosts discriminate intruders from nestmates [9,10] and accomplish that mainly via chemical recognition, using cues encoded in the outermost layer of the cuticle; a complex hydrocarbon blend (typically, alkanes, alkenes, and branched hydrocarbons) which works as a colonial chemical signature [11,12,13,14,15,16]. Colony members share the same signature (same hydrocarbons in similar proportions, e.g., [17], but see [18]); within a species, members of different colonies typically have distinctive colony signatures (same hydrocarbons in different proportions), whereas qualitative differences (i.e., different hydrocarbons) distinguish species [19]. Typically, colony residents perceive and respond aggressively to qualitative differences in the cuticular hydrocarbon blends of approaching individuals (e.g., non-nestmates of another species, U-present model [19]), but they also attack those bearing small quantitative differences (e.g., conspecific non-nestmates [20,21,22]). The detection of intruders is effective because social insects perceive very small concentrations of cuticular hydrocarbons [23]—although they need a certain concentration difference to perform the discrimination [24] and a certain overall concentration of hydrocarbons to respond aggressively to intruders [25,26]. Newly emerged non-nestmates are tolerated in alien colonies because they are poor in hydrocarbons (often both in total amount and in the number of individual compounds, e.g., [27,28]), whereas individuals bearing a sufficiently high concentration of cuticular hydrocarbons are typically detected and rejected if their blend does not match the colony signature [29,30,31]. Therefore, differences in (1) the composition of the blend of cuticular hydrocarbons (qualitative differences); (2) the relative proportions of the cuticular hydrocarbons (quantitative differences) and (3) the concentration (absolute amount) of cuticular hydrocarbons are all relevant variables in the detection of intruders by colony residents.

Social parasites evade host detection by means of one or more chemically deceptive strategies that minimize such differences. Some social parasites exhibit chemical mimicry; their cuticular chemical profile matches that of their host both qualitatively and quantitatively making it difficult for hosts to differentiate parasites from nestmates [28,32,33,34]. For example, the cuticular chemical profile of *Myrmica karavajevi* queens, which are rare, workerless, social parasites of *Myrmica*
*scabrinodis* ants, overlaps that of its host—and more precisely, that of host queens (as opposed to that of host workers), allowing parasites to live undisturbed at the host colony and achieve a high social status [35]. Similarly, among myrmecophile staphylinid rove-beetles that live at *Eciton* army-ant colonies, the more accurate the cuticular hydrocarbon mimics of the hosts, the more socially integrated the beetles [36].

Other social parasites circumvent host detection by bearing cuticular hydrocarbons in traces or low concentrations or expressing mainly hydrocarbons which are not meaningful as recognition cues; any of these strategies makes it difficult for hosts to detect non-nestmates; parasites are supposedly below or close to the host perception threshold for recognition cues (chemical insignificance, [28,37]). For example, *Megalomyrmex symmetochus* ants are social parasites of the fungus-growing ant *Sericomyrmex amabilis*. The parasite queen produces her own workers before producing sexuals, and parasite workers have chemical profiles qualitatively different from those of host workers, but significantly poorer in hydrocarbons and evade host detection by chemical insignificance [38].

In some cases, social parasites are both chemically mimetic and chemically insignificant and express these traits either sequentially or at the same time. For example, the queens of the social parasite ant *Polyergus rufescens* have only traces of cuticular hydrocarbons before host nest invasion, whereas they exhibit higher hydrocarbon concentrations (and a host-matching signature) once established at the host nest [39]. This suggests that lacking cuticular hydrocarbons may be advantageous to infiltrate host colonies but disadvantageous in the long term (see also [40]). In contrast, the queens of the social parasites *Acromyrmex insinuator* ants, as well as *Polistes atrimandibularis* and *Polistes semenowi* social parasite wasps, maintain low cuticular hydrocarbon concentrations (i.e., full chemical insignificance) also as established parasite queens [41,42,43,44].

If beneficial in terms of making parasites difficult to detect, chemical insignificance may have consequences in terms of overall water balance. Indeed, the outermost cuticular layer of hydrocarbons is thought to have evolved primarily as a barrier against desiccation [45,46,47]. Insect water loss via the cuticle (incl. cuticular and respiratory loss) is substantial ([48]; >70% of body water [49]) and the cuticular hydrocarbon layer has a major role in reducing water permeability in the cuticle, estimated at up to 1300% reduction [50,51,52]. An effective cuticular hydrocarbon layer is therefore vital.

Social parasites share the same environment as their hosts—they live together at exactly the same nest—so that both are exposed to the same microclimatic conditions. However, chemically insignificant parasites are not as protected as their hosts against dehydration, as they have a weaker hydrophobic barrier [37]. If cuticular hydrocarbons limit water loss, what specific adaptation have social parasites evolved to stay chemically insignificant all along the colony cycle without incurring dehydration? 

Here, I tested whether chemical insignificant parasites exhibit behavioral adaptations that limit water loss.

*Polistes* paper wasps represent a particularly interesting model to test the potential costs of a reduced cuticular layer of hydrocarbons in terms of dehydration risk. Unlike ants, whose nests are typically underground, paper wasp nests are suspended from relatively aerial structures (porches, roofs, plants, etc.) and are open, lacking comb-protecting envelopes [53]. This makes it easy to perform behavioral observation [54] but exposes combs and adult wasps to overheating during summer. *Polistes* wasps actively cool their nests by fanning their wings and building extra cells they leave empty [55,56,57], but how do they protect themselves? Heat stress induces desiccation [58]. Therefore, it is surprising that paper wasps’ social parasites can afford months-long chemical insignificance without paying costs in terms of water balance. Indeed, both *Polistes atrimandibularis* and *Polistes semenowi* have poorer cuticular chemical profiles than their hosts [33,43,44]. These obligate, workerless, *Polistes* social parasites are phylogenetically relatively close to their hosts [59]. Their general morphology and their body size are similar to their hosts (both foundresses and workers, since paper wasps have no caste dimorphism [53]) and morphological differences are limited to specific body parts, such as the head, first femur, and posterior tibia [60].

Females of *P. atrimandibularis* invade *P. biglumis* host nests in June, before the emergence of the host brood [61,62]. Unlike the related species *Polistes sulcifer*, whose invasion is aggressive [63], *P. atrimandibularis* parasites are host-queen tolerant, meaning that they adopt a non-violent strategy of invasion and cohabit with the single host foundress [64], a strategy employed also by ant social parasites (e.g., [65]). Chemical analyses have shown that before host-nest invasion they have a cuticular signature 1) different from that of the hosts [66] and 2) exceptionally poor in hydrocarbons: just 20% the amount of their hosts [40]. Most surprisingly, parasites stay chemically insignificant until the end of the colony cycle, when brood rearing ends [43], thus exposing themselves to dehydration risk all along summer and colony cycle. After host nest invasion, parasite chemical signature progressively switches to match that of the host through massive qualitative changes [66], but without any increase in the total amount of hydrocarbons [43]. In fact, parasites lose linear alkanes and alkenes in favor of longer-chain, branched hydrocarbons [40]—but the overall effect of these changes on the waterproofing capacity of the cuticle is unknown. While losing alkenes in favor of long-chain hydrocarbons would increase waterproofing, decreasing linear alkanes in favor of branched alkanes would diminish it [16,51,52].

Here, I tested whether the observed low amount of cuticular hydrocarbons in *P. atrimandibularis* social parasites might be counterbalanced by parasites exhibiting behaviors that limit water loss. I compared the behavior of parasites with that of their single host foundresses because parasites and hosts share their colony and therefore are exposed to the same microclimatic condition. Hosts and parasites are phylogenetically related and morphologically very similar but parasites have a lower amount of cuticular hydrocarbons than their hosts. I predicted that, if being active, foraging, and resting at the sunny front side of the nest increases dehydration-stress risk, parasites should be less active, forage less, and rest behind the nest more than their host foundresses. I also predicted that, if being involved in trophallactic exchanges as receivers contribute to limiting dehydration, parasites should act as receivers more often than as donors compared with the host foundresses. I tested these predictions using data from field behavioral recordings. I acknowledge that this will offer only correlative evidence, if any, and that experimental data are needed for direct evidence. However, social parasites are very rare [60,67,68,69,70], and *P. atrimandibularis* parasites are especially rare, as their main host (*P. biglumis*) has a small population size [62]. Therefore, conducting experimental studies to test dry/heat resistance in such parasites would provide evidence for the costs of chemical insignificance but would also put at risk the conservation of social-parasite populations [71,72,73]. Hence, I reverted to analyzing behavioral data from field observations to test the hypothesis that chemically insignificant parasites exhibit behavioral adaptations contributing to preserving themselves from dehydration risk.

## 2. Materials and Methods

### 2.1. Study System

#### 2.1.1. The Host

*Polistes biglumis* is a non-parasitic, boreo-mountain species [74]. In the Alps, populations are small and geographically isolated (although gene flow occurs between close populations [75]). Colonies are founded by a single foundress in South-facing prairies on gentle mountain slopes at 1200–2000 m. a.s.l. in May-early June [62] (Figure 1). The first brood emerges in July and the nest is abandoned less than two months later (end of August-early September) [61]. The colony cycle is therefore very short due to harsh climatic conditions which are reflected into extreme microclimatic variations at the nest, where temperatures in June-July-August may vary between 5 °C (typically, during the night and bad weather) to highs of >40 °C (midday hours in sunny days) [76]. 

#### 2.1.2. The Parasite

The parasite *Polistes atrimandibularis* is a workerless, obligatory, permanent social parasite distributed around the Mediterranean and Caspian basins [77]. Although parasite to several *Polistes* species [78], the only host in the Alps is *P. biglumis* [62]. Parasite prevalence is generally low, but locally relatively high (up to ~24% parasitized nests) and causes local adaptations in host life-history traits [62], a common outcome of the coevolutionary arms race between host and parasite [79,80,81,82].

The parasite female enters the host colony before host brood emergence, that is, when the single host foundress is the only adult at the nest. The parasite subdues the host foundress using a peaceful invasion tactic [64] and forces the host to work and rear her brood [61]. The host queen’s reproductive success is low in parasitized nests (although parasites protect host colonies from predators [62]) and host daughters largely turn into working for the parasite and rearing her offspring (host daughters are largely destined to become future foundresses in free-living colonies [83]).

### 2.2. Data Collection

I used videotaping data collected from 45 colonies of *Polistes biglumis* parasitized by *P. atrimandibularis* in 2006 (15 June–18 August) and 2007 (23 June–1 August) using a Canon MV960 camcorder in two different populations (one in the Alps: Montgenèvre, Hautes Alpes, France, elevation: 1860 m, and one in the Apennine: Monte Mare, Isernia, Italy, elevation: 1740 m). Resident wasps were individually marked with enamel paint. The camera was placed at approximately 20 cm from the front face of the comb. Nests are built close to the ground, attached to the vertical surface of rocks on well-lighted sites and cells have horizontal orientation [61] (Figure 1), making videotaping of the comb front face especially easy. The behavior of colony residents (hosts and parasites) was recorded in 105 observation sessions (1-6 observation sessions per colony; 104.4 ± 21.1 min per colony, average ± s.e.). In total, data come from 89 h of videotaping performed during sunny days between 9:00–17:00 h.

I measured the proportion of time each social parasite and its single (and subdue) host foundress spent in the following behaviors: 

Activity: any behavior (except resting);Time at the nest (vs outside the nest, presumably foraging for nectar or preys);Resting at the front face of the nest (where cells open) *versus* resting on the backside of the nest—typically within a wasp-body-size distance from the nest stalk (petiole) that suspends the comb to the substrate; the back face of the nest is shadowy, whereas the front face is sunny for a large part of the day;Receiving *versus* offering liquids during trophallaxis (regurgitated liquid transfer between individuals) with any other adults or with larvae.

Each nest contributed data only from the single parasite female and the single subdue host foundress. Data were analyzed with generalized linear mixed models (GLMM, function glmer, package *lme4* [84]; R version 1.3.959 [85]) with a binomial error structure (logit link) and with the duration of behaviors (namely, activity vs rest; time at the nest vs outside the nest; resting at the front nest face vs. behind the nest) as the dependent variable. Data on trophallactic exchanges were count data (number of liquid donations offered and received to and from any member of the colony) and were analyzed with a GLMM for Poisson distributed data. In all models, I tested for the effect of who was the actor (either the social parasite or the host foundress). Since populations differ in life-history traits and behavior [62,86], I controlled for population, time of the day (three levels: morning: 9:00–11:59 h; midday: 12:00–14:00; afternoon: 15:00–17:00,) and period during the colony cycle (two levels: before or after the emergence of parasite and/or host offspring in the nest). The interactions actor*population and actor*time of the day were included in preliminary models to account for potential differences in the relative behavior of parasites and hosts between populations or daily schedules. Colony ID was entered as a random factor, given that observations of parasites and hosts from the same nest were not independent (the behavior of each foundress is here control for that of her own parasite, to account for colony-specific features, such as the number of larvae in the nest and/or the microclimatic condition) [87]. If needed, a case-level random factor was added to correct for overdispersion.

The following predictors: population, time of the day, and period during the colony cycle, entered in the preliminary models, were removed from subsequent models when they were non-significant after comparing the competing models’ AIC values; results are reported from the models with the lowest AIC (Akaike Information Criterion) value to avoid information loss.

In the figures, data are represented as percentages (i.e., percentage of activity and of time at the nest on total observation time; percentage of time resting at the front face of the nest on total time resting); data on trophallactic exchanges are shown using boxplots.

## 3. Results

Parasites were significantly less active than their respective host foundresses and both were significantly more active during morning and midday hours than during the afternoon (GLMM, [parasite]: β = −1.494 ± 0.462, Wald χ^2^ = 10.452, df = 1, *p* = 0.001; time of the day: χ^2^ = 11.198, df = 2, *p* = 0.004) (Figure 2). 

Parasites spent a larger proportion of time at the nest than their host foundresses (GLMM, [parasite]: β = 3.815 ± 0.850, χ^2^ = 20.148, df = 1, *p* < 0.0001) (Figure 3). 

Intriguingly, when they were at the nest, parasites spent proportionally less time than their host foundresses resting at the sunny front face of the nest than at the shadowy back face ([parasite]: β = −2.554 ± 0.880, χ^2^ = 8.425, df = 1, *p* = 0.004) (Figure 4). 

Finally, although parasites were rarely involved in trophallactic exchanges, they offered liquid drops to their host foundresses significantly less often than host foundresses did towards their parasites ([parasite]: β = −2.554 ± 0.880, χ^2^ = 8.4253, df = 1, *p* = 0.004) (Figure 5). 

## 4. Discussion

These results suggest that strongly chemically insignificant social parasites, which bear a significantly smaller amount of cuticular hydrocarbons than their hosts, adjust their behavior in ways that are compatible with a limiting water balance. Although parasites cohabit with host foundresses at the host nest, they were less active than their own host foundresses, spent more time at the nest, and rested less at the sunny front face of the nest than their host foundress, a result which contradicted expectations for dominant females. Finally, in the trophallactic exchanges occurring at the colony, parasites received proportionally more liquid drops (and offered fewer) than their host foundresses. These results were consistent between two geographically isolated populations located at a distance larger than 600 km and characterized by significant climatic differences [83].

Typically, variations in the amount of cuticular hydrocarbons have been identified as adaptive responses to environmental conditions; for instance, variations have been reported by age and caste/task [88,89,90,91,92,93,94]. Typically, workers engaged in outward activities have larger amounts of cuticular hydrocarbons than those working at the nest [95,96]. Similarly, the experimental introduction of non-nestmates caused an increase in the amount of cuticular hydrocarbons in residents, possibly boosting colony-identity cues [97]. If these works have shown that age, tasks, and social variations result in variation in cuticular hydrocarbon concentration, the low concentration of hydrocarbons in *P. atrimandibularis* parasites might be the cause, rather than the consequence, of the behavioral change. 

It may be argued that some of the current results were expected irrespective of water balance constraints. Indeed, if social parasites acquire the highest rank in the colony, they are expected to behave as dominant females do: little activity, little foraging, and more trophallactic drops received than offered [53,60,98]. However, *P. atrimandibularis* is a rather atypical social parasite in that the parasite female contributes to forage for protein and feed the brood. This occurs via a peculiar predation strategy where parasites prey on the brood of *P. biglumis* colonies other than the one they have established in [99]. However, although active foragers, parasites leave the nest less than their host foundress; even if driven mainly by dominance hierarchy rules, limiting activity outside the nest would nonetheless act as a water-saving strategy.

Social parasites usually achieve the dominant position at the host colony and both adult parasites and their brood get priority on hosts. For instance, the larvae of the social parasite wasp *Polistes sulcifer* are cared by host workers more than host brood [100]. Among ants, *Myrmica schencki* workers protect queen-destined pupae more than worker pupae in case of nest disturbances [101] and exhibit higher protection responses towards queens than workers; when colonies are parasitized by *Maculinea rebeli* butterflies, parasite offspring deserve the same level of protection as host queens [102].

Dominant individuals also usually occupy central positions within the group in many social animals [103]. Similarly, the use of nest space is typically structured in social insects; queens occupy the central nest positions in bumblebees, ants, and wasps ([104,105,106,107], respectively). Spatial segregation applies also to *Polistes* wasps; the most dominant female rests at the front face of the comb, rather than behind the nest [108,109]. A recent analysis confirmed that and highlighted that dominant wasps occupy the center of the front face [110]. In contrast, the current results showed that *P. atrimandibularis* rested behind the nest, in the shadow, rather than at the sunny front face. *Polistes* social parasites do acquire the dominant position as they subdue the host foundresses [60] and acquire the chemical profile of dominant host females [111]. In particular, *P. atrimandibularis* social parasites inhibit host foundress fecundity and stop host brood production [112] and even manipulate hosts into working by means of behavioral dominance [86]. The current data suggest that parasites stayed on the back of the nest irrespective of their reproductive dominance at the host colony; indirectly, they support the hypothesis that limiting dehydration can influence parasite behavior more strongly than dominance rank.

Desiccation stress may be countered via different physiological adaptations besides those involving variations in cuticular hydrocarbon concentrations or composition. In *Drosophila,* at least three physiological mechanisms have been identified via experimental-evolution experiments in xeric environments: fruit flies enhance baseline water content in their body, lessen water loss rates (e.g., via tighter control of respiratory and excretory water loss), and/or enhance dehydration tolerance (i.e., tolerate lower percentage body water) [113]. These mechanisms may parallel plastic adjustments of cuticular hydrocarbon blends under desiccation stress (i.e., acclimation changes predicted on the base of the biophysical properties of the different hydrocarbon classes [114]). However, chemical insignificant social parasites exhibit chemical profiles whose characteristics seem inappropriate to the microclimatic conditions where they live but multiple physiological water-saving mechanisms may help explain how they survive and why, sometimes, the concentration of hydrocarbons is uncorrelated to xeric conditions (e.g., [115]). Additionally, a thick cuticle, as reported in *Polistes* social parasites, may contribute to diminishing water loss by transpiration [60]. 

The behavioral mechanisms addressed in this work add up to the adaptations which might promote resistance to desiccation stress in chemically insignificant social parasites. The question that opens now is how chemically insignificant social parasites manage to achieve and maintain their dominant position in the colony social hierarchy while bearing a chemically invisible cloak and, in the case of *P. atrimandibularis*, while also avoiding the central nest position in a spatially structured society.

## 5. Conclusions

In conclusion, behavioral data from field observations suggest that chemically insignificant social parasites have evolved behavioral water-saving strategies that allow them to cope with the low amount of cuticular hydrocarbons. On one side, the low concentration of cuticular hydrocarbons may allow social parasites to evade host detection, but, on the other, it exposes them to desiccation stress. Behaviors such as being less active than hosts, spending more time at the nest, and resting behind the nest, rather than at the sunny front face, may contribute to limiting dehydration risk in these social parasites.

## Figures and Tables

**Figure 1 insects-12-01006-f001:**
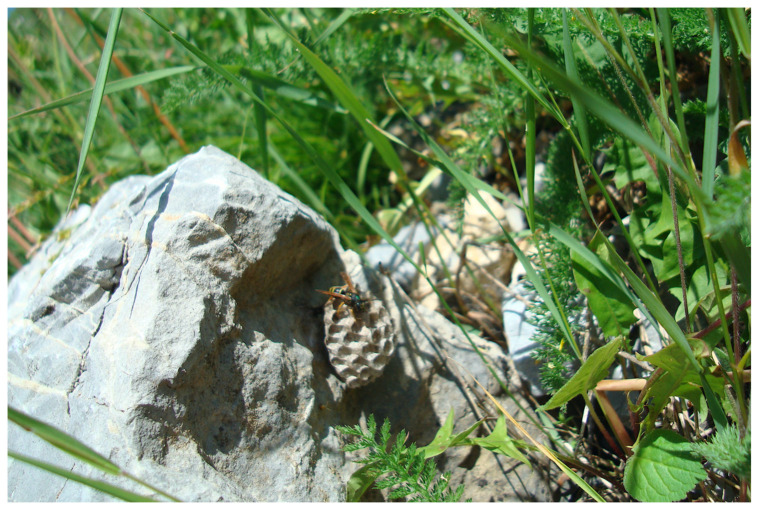
A *Polistes biglumis* nest built on a rock close to the ground.

**Figure 2 insects-12-01006-f002:**
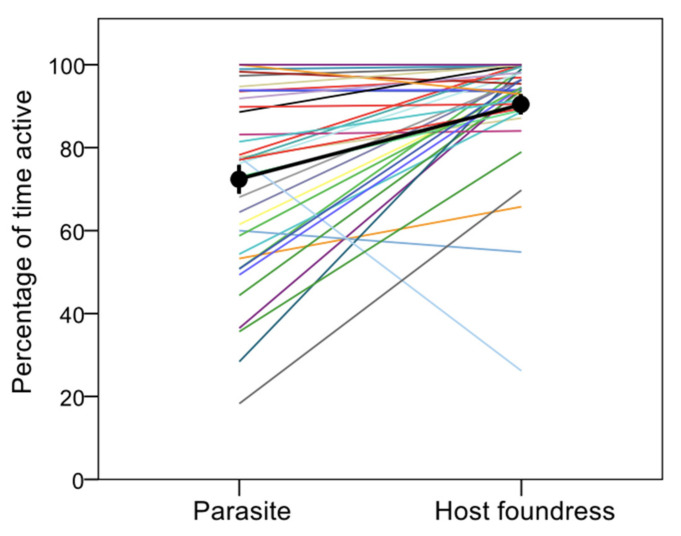
Percentage of time parasites and hosts were active (vs resting) during the observation session. Each line links the values for the parasite (left) and her host foundress (right) living at the same colony (pairwise data); the black thick line links the mean values (± s.e.) for parasites and foundresses (after averaging by colony). Variation as a function of time of the day is not shown.

**Figure 3 insects-12-01006-f003:**
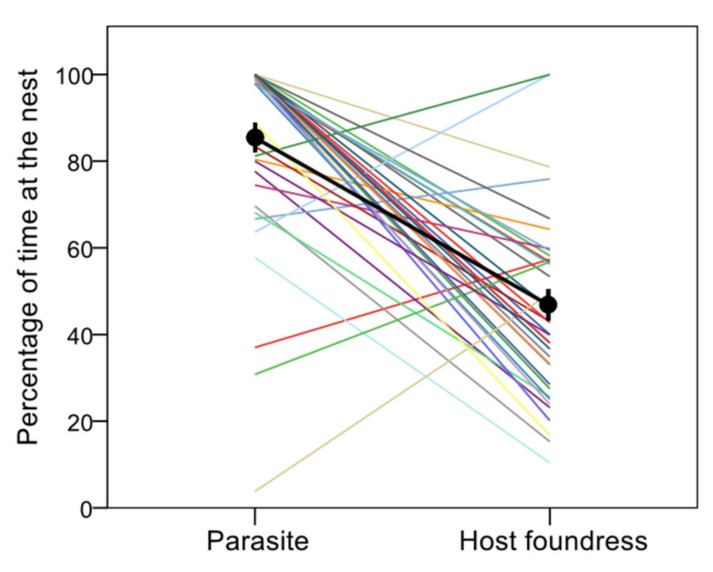
Percentage of time parasites and hosts spent at the nest. Each line links the values for the parasite (left) and her host foundress (right) living at the same colony (pairwise data); the black thick line links the mean values (± s.e.) for parasites and foundresses (after averaging by colony).

**Figure 4 insects-12-01006-f004:**
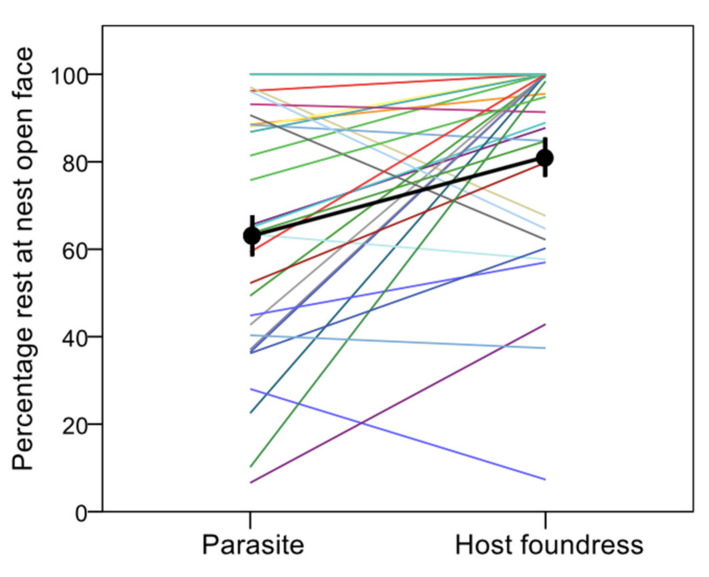
Percentage of time parasites and hosts spent resting at the front face of the nest. Each line links the values for the parasite (left) and her host foundress (right) living at the same colony (pairwise data); the black thick line links the mean values (± s.e.) for parasites and foundresses (after averaging by colony).

**Figure 5 insects-12-01006-f005:**
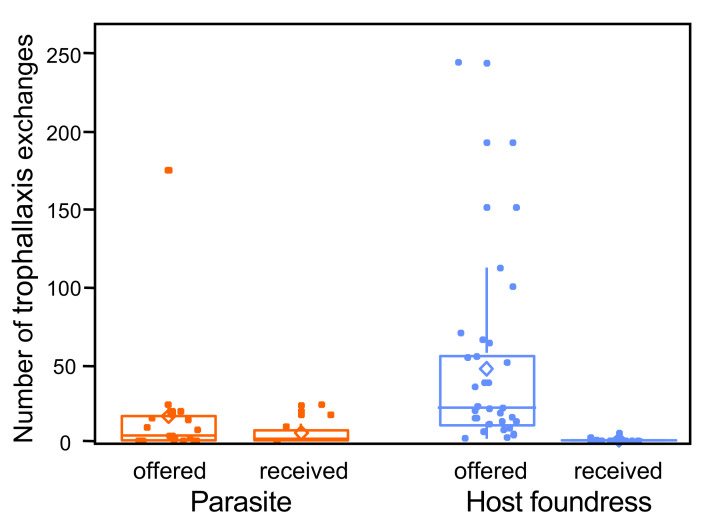
Number of trophallactic exchanges offered and received by parasites and hosts. The boxplots show medians, quartiles, 5th and 95th percentiles, and minimum and maximum values outside of the percentiles (color dots); diamonds represent means; points are jittered to prevent overlap.

## Data Availability

The data used in this work are available in Appendix A.

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
