# Peer review of "Chemically Insignificant Social Parasites Exhibit More Anti-Dehydration Behaviors than Their Hosts"

_insects, 2021, doi:10.3390/insects12111006_

Round 1

Reviewer 1 Report

The manuscript "Chemically insignificant social parasites exhibit more anti-dehydration behaviors than their hosts" describes how Polistes parasites that share the nest with a Polistes host may adjust their behaviour to avoid desiccation due to reduced hydrocarbons (likely an adaptation to avoid being rejected by the host) compared to the hosts. 

The whole manuscript was well-written, and interesting. The methods and analyses were appropriate to begin investigating behavioural differences among the host and parasite queens on nests (n = 45 colonies) in this species. The results were clearly presented and the conclusions were accurately drawn from the results. Overall, I really enjoyed this manuscript, and look forward to the sequel. 

I only found a few minor aspects that the authors might want to consider revising. 

  • Abstract: Genus & species are not italicized.
  • Line 85: "strategy" should be "strategies"
  • Line 118: "parasite" should be plural
  • Line 135: "They" should be "Their"
  • Line 148: "chemical" should be "chemically"
  • Lines 160-1: It's not clear where this sentence belongs.
  • Lines 227-30: "Nests are built close to the ground..." consider including an image of the nest to illustrate this.
  • Line 272 & Figure 4: Consider using a box plot instead of mean +/- se, since the data are not normally distributed. 
  • Line 372: "explaining" should be "explain"

Reviewer 2 Report

Dear author,

I revised your ms, which deals with a very intriguing question: how does a social parasite using chemical insignificance (thus with reduced CHCs) cope with the increased risk of water loss?

While the manuscript is overall well readable, the topic nice and it surely deserves attention, I am sorry to say that I am not convinced by your manuscript. My opinion is that you do not provide solid ground for your conclusions. Below I specify the main weaknesses that ms present in my opinion. I think additional experiments and/or observation would be needed to be more confident in concluding that these behavioural differences exists and are adaptation evolved to counteract water loss due to the chemical insignificant startegy.

1) experimental design: lack of proper control.

I do not believe that the subdue host queen, in the same usurped colony, is the right (and unique) control here. It is surely useful to have this contrast, but another needed control would have been a queen on her queenright, not usurped nest. This is the usual control group when comparing a social parasite with its social counterpart and might be informative of the usual/normal level of target behaviours in normal conditions. Here, indeed, the problem is that the usurped queen might have her behaviour altered by the presence of the social parasite. I would add as control queenright queen, both without and with workers (so that dominance and foraging needs factors could be taken into account)

2) interpretation of the behavioural differences as adaptations for dehydration. You consider as indicative of adaptation against dehydration due to low CHC amount:

  • reduced activity
  • bias in position on the nest (behind rather than on frontal part)
  • less foraging
  • bias toward receiving than giving trophallaxis

While I see how each of this items could help in reducing water loss, this is not the unique interpretation. Actually, all of them are quite consistent with the "social parasite" syndrome/behaviour: dominant exploiters of a social host.

For a social parasite it pays to avoid foraging (or spending less time foraging), resting, and receive sugary liquid food, even if this is not related to water loss. I am not saying it is not, simply that there are other possible explanations (energy related, dominance related, simply lifestyle related) which cannot be discarded. You argument a little bit this , at lines 325 and following, but in my opinion this is not sufficient to then orient the interpretation toward these behaviours being adaptations against water loss. Here, proper experiments and/or correlational data are needed to disentangle the several possible explanations. For example: a) young individuals do not have immediately a full chemical profile...hod do they behave?; b) are these behaviours influenced by dominance level of the social parasite? c) is the magnitude and direction of these behavioural differences  influenced by climate (e..g humidity etc)?

Also, I am wondering whether other social parasites with no reduced CHCs show nonetheless these behaviour. If possible to be done, this could be a revealing experiment.

My point here is that the correlational data you present might suggest strategies against water loss, but not only, and while you suggest something in the discussion the ms is too tailored on the water loss hypothesis. I do not think this is proven in your manuscript.

3) Aspects which have not been taken into account but might be really significant for the interpretation.

It seems to me you have data along a somehow long period, from June to August.  Is there any effect of timing? this might allow you to differentiate between a dominance hp and a water loss hp. Both factor might change over time (temperature and humidity change, and also dominance and integration into the colony changes, form the very first days to the last days...) and not necessarily i the same direction. Thus, understanding IF and HOW these behaviours (foraging, trophallaxis, activity etc) change over time might allow you to better understand the related advantages.

4) General framework and wording

Overall, as explained above, I think the data are interesting but not unequivocally supporting your claims, as on the contrary your framework and wording suggest.

For example, you say the ms wants to test (lines 120, 162). I would say "investigate", or "explore", as you predictions are not so specific to allow a TEST of the hypotheses, as alternative explanations might be proposed and as the data are purely correlative. Also, at lines 359,361; 383-385 the unique interpretation given is that of the water loss balance, which to me is a speculation. 

5) Parasite rarity and the possibility to do experiments.

I see the concern here, and I appreciate the care about this aspect. However, I also think that if a model system does not allow a proper and sound test, not necessarily the test should be carried out. I mean, you coudl gather these important data, and simply explain them in light of several alternative hypotheses, suggesting your preferred one (and why) but leaving open the interpretation to the reader (maybe also suggesting what could be done if more parasite were available). in this way you would simply report and discuss the scientific evidence. 

I am sorry to disappoint on this occasion with my non-positive evaluation. I really think the topic is nice, and surely there has been significant work behind. However, I do not think you present enough and unequivocal evidence for your claims.

Reviewer 3 Report

Chemically insignificant social parasites exhibit more anti-dehydration behaviors than their hosts

Maria Cristina Lorenzi

This is a quite interesting paper which combines Polistine social and parasitic behavior with thermal biology. Due to the very well written text I found only a few minor comments for improvement. The statistics is quite thoroughly done. Therefore the interpretation (see MS title) seems plausible. Of course there remain open questions but this is addressed quite well in the MS.

Therefore, I suggest acceptance with minor revision as indicated in my comments below.

Comments:

L10: insert long dash also after “hydrocarbons”

L105: “… making a parasite difficult to …” or “making parasites difficult to …”

L135: “Their general …”

L147: “…and are exceptional …

L163: delete comma

L164-169: This sentence is really hard to read and therefore has to be revised. It is way too long and complicated. Also: comma not needed after “sunny”. Please edit!

L183: insert paragraph space

L213-220: This whole paragraph should be omitted. It is just a repetition of what was already said in the introduction. This information is out of place in the methods section.

Fig. 2: delete one “;” in legend

Reviewer 4 Report

The author presents a nice study on behavioural adaptations that may help a social parasite wasp to balance the adavantages to be chemically insignificant with the risk of dessication compared with the host wasp. The study is well done, methods are adequate, results well presented and discussion well focussed on the results. I have made only a couple of comments directly on the ms (see attached pdf file). These are just suggestions, so not really changes necessary to accept the ms. Overall, a great study.

Round 2

Reviewer 2 Report

Dear author,

I am quite puzzled by the fact that you did not take into account most (almost none) of my comments.

I am still convinced that your data do not provide evidence about what you claim. The hp of anti-dehydration behaviours is one of the several (and some other more likely) one can propose. the experimental design is not appropriate, as it does not allow to disentangle the several hp. I report below my answers to your response letter. Despite I think the ms could bring interesting results, I believe that how it is frame it is rather misleading in its one-sided interpretation and not presenting enough and strong evidence. 

best regards

REVIEWER 2

 Dear author,

I revised your ms, which deals with a very intriguing question: how does a social parasite using chemical insignificance (thus with reduced CHCs) cope with the increased risk of water loss?

While the manuscript is overall well readable, the topic nice and it surely deserves attention, I am sorry to say that I am not convinced by your manuscript. My opinion is that you do not provide solid ground for your conclusions. Below I specify the main weaknesses that ms present in my opinion. I think additional experiments and/or observation would be needed to be more confident in concluding that these behavioural differences exists and are adaptation evolved to counteract water loss due to the chemical insignificant startegy.

AS THE REVISION DID NOT ADD DATA NOR EXPERIMENT, THIS CRITICS STILL APPLIES

  • experimental design: lack of proper

I do not believe that the subdue host queen, in the same usurped colony, is the right (and unique) control here. It is surely useful to have this contrast, but another needed control would have been a queen on her queenright, not usurped nest. This is the usual control group when comparing a social parasite with its social counterpart and might be informative of the usual/normal level of target behaviours in normal conditions. Here, indeed, the problem is that the usurped queen might have her behaviour altered by the presence of the social parasite. I would add as control queenright queen, both without and with workers (so that dominance and foraging needs factors could be taken into account)

REPLY: I reasoned about using queens from un-parasitized nests as controls. However, I concluded that I would have failed to control for the nest microclimatic conditions, which are likely to affect thermoregulation and waterproofing of adult wasps directly. Indeed, many of your comments below point to the precise weather conditions and you ask how they affect the wasp behavior (you mention humidity and temperature, for example). This is why collecting data from parasite and host foundress sharing the same nest and doing that exactly at the same day and time of the day was important: this way, the effect of temperature or humidity is the same on both parasite and foundress.

I SEE THE POINT, BUT BY TRYING TO CONTROL FOR MICROCLIMATE CONDITIONS YOU ACTUALLY INTRODUCE AN EVEN BIGGER DIFFERENCE: DOMINANCE STATUS. IT IS WELL KNOWN THAT DOMINANCE INFLUENCES BEHAVIOURS, ESPECIALLY THOS STUDIED IN THIS STUDY, SO COMPARING ONLY SUBORDAINTE FOUNDRESSES AND DOMINANT SOCIAL PARASITES IS NOT CORRECT. WITHOUTH A COMPARISON WITH DOMINANT QUEENRIGHT QUEENS, YOUR STUDY SIMPLY DEMONSTRATE DIFFERENCE BETWEEN TWOP INDIVIDUALS WHICH ALSO HAVE A DIFFERENT DOMINANT STATUS.

In addition, the number and size of the larvae would differ between nests; even adding the number of larvae as a covariate will fail to control for larval hunger condition and for larval developmental phase.

THIS COUDL CLEARLY HAVE AN IMPACT, BUT IF THIS WAS A REASON TO NOT PERFORM COMPARISON MUCH OF THE RESEARCH ON SOCIAL INSECTS WOULD NOT HAVE BEEN DONE! ADDING QUEENRIGHT CONTROL COLONIES (IN ADDITION TO SAME NEST FOUNDRESSES) WOULD HAVE ALLOWED TO INFER THE IMPORTANCE OF AMONG COLONIES DIFFERENCES.

  • interpretation of the behavioural differences as adaptations for dehydration. You consider as indicative of adaptation against dehydration due to low CHC amount:
  • reduced activity
  • bias in position on the nest (behind rather than on frontal part)
  • less foraging
  • bias toward receiving than giving trophallaxis

While I see how each of this items could help in reducing water loss, this is not the unique interpretation. Actually, all of them are quite consistent with the "social parasite" syndrome/behaviour: dominant exploiters of a social host.

REPLY: I acknowledged that at lines 338-343.

IN MY OPINION ADDING A FEW SENTENCES IS NOT ENOUG IF THEN THE ENTIRE PAPER (TITLE ABSTRACT ETC) IS PRESENT THE PREFERRED HP AS THE MAIN AND MOST LIKELY INTERPRETATION OF THE RESULTS.

For a social parasite it pays to avoid foraging (or spending less time foraging), resting, and receive sugary liquid food, even if this is not related to water loss. I am not saying it is not, simply that there are other possible explanations (energy related, dominance related, simply lifestyle related) which cannot be discarded. You argument a little bit this , at lines 325 and following, but in my opinion this is not sufficient to then orient the interpretation toward these behaviours being adaptations against water loss. Here, proper experiments and/or correlational data are needed to disentangle the several possible explanations. For example: a) young individuals do not have immediately a full chemical profile...hod do they behave?; b) are these behaviours influenced by dominance level of the social parasite? c) is the magnitude and direction of these behavioural differences influenced by climate (e..g humidity etc)?

REPLY: these interesting questions might be addresses in future studies.

THESE ARE ACTUALLY EXPERIMENTS THAT ARE NEEDED HERE, TO DISENTANGLE THE SEVERAL HP AND SUPPORT WHAT THE MS CLAIMS TO HAVE FOUND. I AM ACTUALLY SUGGESTING WAYS TO PROVE THE MS HP, NOT ADDITIONAL EXPERIMENT TO BE DONE IN THE FUTURE

Also, I am wondering whether other social parasites with no reduced CHCs show nonetheless these behaviour. If possible to be done, this could be a revealing experiment.

REPLY: this interesting question might be addresses in future studies.

AS ABOVE, THIS IS SOMETHING I THINK IT IS NEEDED IF YOU WANT TO PRESENT AN INTERESTING HP (BUT ONE OF MANY) AS SOMETHING PROVED (YOU TALK ABOUT A TEST IN THE ABSTRACT)

My point here is that the correlational data you present might suggest strategies against water loss, but not only, and while you suggest something in the discussion the ms is too tailored on the water loss hypothesis. I do not think this is proven in your manuscript.

  • Aspects which have not been taken into account but might be really significant for the

It sems to me you have data along a somehow long period, from June to August. Is there any effect of timing? this might allow you to differentiate between a dominance hp and a water loss hp.

REPLY: Time of the day and period during the colony cycle were entered in the analyses.

YES, I SAW THAT, BUT THEY ARE NOT REALLY ANALYSED (TIME OF THE DAY IS, TIME FROM USURPATION OR TIME IN THE SEASON IS NOT) NOR I CAN SEE GRAPHS. YOU ARE BASICALLY CONTROLLING FOR THESE VARIABLES, WHILE WHAT I AM SUGGESTING IS TO USE THESE FACTORS TO TRY TO DISENTANGLE THE POSSIBLE HPs YOU HAVE (E.G. DOMINANCE AND WATER LOSS)

Both factor might change over time (temperature and humidity change, and also dominance and integration into the colony changes, form the very first days to the last days...) and not necessarily i the same direction. Thus, understanding IF and HOW these behaviours (foraging, trophallaxis, activity etc) change over time might allow you to better understand the related advantages.

REPLY: these questions might be addresses in future studies.

AS ABOVE, I THINK THESE ARE QUESTIONS THAT SHOULD BE ADDRESSED HERE.

  • General framework and wording

Overall, as explained above, I think the data are interesting but not unequivocally supporting your claims, as on the contrary your framework and wording suggest.

For example, you say the ms wants to test (lines 120, 162). I would say "investigate", or "explore", as you predictions are not so specific to allow a TEST of the hypotheses, as alternative explanations might be proposed and as the data are purely correlative. Also, at lines 359,361; 383-385 the unique interpretation given is that of the water loss balance, which to me is a speculation.

REPLY: I acknowledged alternative explanations at lines 338-343.

AS ABOVE, ONLY A FEW LINES IN MY OPINION DO NOT CHANGE THE OVERAL FRAMEWORK OF THE MS WHICH STILL PRESENT WATER LOSS AS THE MAIN HP AND ALMOST PROVEN FACT...IN MY OPINION IT IS NOT.

  • Parasite rarity and the possibility to do

I see the concern here, and I appreciate the care about this aspect. However, I also think that if a model system does not allow a proper and sound test, not necessarily the test should be carried out. I mean, you coudl gather these important data, and simply explain them in light of several alternative hypotheses, suggesting your preferred one (and why) but leaving open the interpretation to the reader (maybe also suggesting what could be done if more parasite were available). in this way you would simply report and discuss the scientific evidence.

REPLY: Thanks for the suggestions.

IT SEEMS TO ME THAT THE SUGGESTION HAS NOT BEEN TAKEN INTO CONSIDERATION. I WOULD NOT HAVE ANY ISSUE IN ACCEPTING A MS PROPOSING JUST THE DATA AND THEN REPORT MANY HP AS THE POSSIBLE EXPLANATION, WHILE AS IT STANDS IT IS PRESENTED AS A TEST FOR THE HP OF WATER LOSS: IN MY OPINION THE EXP. DESIGN DOES NOT ALLOW THAT.

I am sorry to disappoint on this occasion with my non-positive evaluation. I really think the topic is nice, and surely there has been significant work behind. However, I do not think you present enough and unequivocal evidence for your claims.

AGAIN, I REALLY THINK THERE IS NICE AND INTERESTING MATERIAL IN THE STUDY, SIMPLY, I DO NOT THINK THE HP OF WATER-LOSS IS PROVEN BY YOUR EXP DESIGN AND DATA.

Author Response

Dear Editor,

I replied to the second round of comments by reviewer 2 by adding a new reply labelled “REPLY 25/10/2021” (in bold) below the reviewer comments.

Best,

Cristina Lorenzi

Dear author,

I am quite puzzled by the fact that you did not take into account most (almost none) of my comments.

I am still convinced that your data do not provide evidence about what you claim. The hp of anti-dehydration behaviours is one of the several (and some other more likely) one can propose. the experimental design is not appropriate, as it does not allow to disentangle the several hp. I report below my answers to your response letter. Despite I think the ms could bring interesting results, I believe that how it is frame it is rather misleading in its one-sided interpretation and not presenting enough and strong evidence. 

best regards

REVIEWER 2

 Dear author,

I revised your ms, which deals with a very intriguing question: how does a social parasite using chemical insignificance (thus with reduced CHCs) cope with the increased risk of water loss?

While the manuscript is overall well readable, the topic nice and it surely deserves attention, I am sorry to say that I am not convinced by your manuscript. My opinion is that you do not provide solid ground for your conclusions. Below I specify the main weaknesses that ms present in my opinion. I think additional experiments and/or observation would be needed to be more confident in concluding that these behavioural differences exists and are adaptation evolved to counteract water loss due to the chemical insignificant startegy.

AS THE REVISION DID NOT ADD DATA NOR EXPERIMENT, THIS CRITICS STILL APPLIES

  • experimental design: lack of proper

I do not believe that the subdue host queen, in the same usurped colony, is the right (and unique) control here. It is surely useful to have this contrast, but another needed control would have been a queen on her queenright, not usurped nest. This is the usual control group when comparing a social parasite with its social counterpart and might be informative of the usual/normal level of target behaviours in normal conditions. Here, indeed, the problem is that the usurped queen might have her behaviour altered by the presence of the social parasite. I would add as control queenright queen, both without and with workers (so that dominance and foraging needs factors could be taken into account)

REPLY: I reasoned about using queens from un-parasitized nests as controls. However, I concluded that I would have failed to control for the nest microclimatic conditions, which are likely to affect thermoregulation and waterproofing of adult wasps directly. Indeed, many of your comments below point to the precise weather conditions and you ask how they affect the wasp behavior (you mention humidity and temperature, for example). This is why collecting data from parasite and host foundress sharing the same nest and doing that exactly at the same day and time of the day was important: this way, the effect of temperature or humidity is the same on both parasite and foundress.

I SEE THE POINT, BUT BY TRYING TO CONTROL FOR MICROCLIMATE CONDITIONS YOU ACTUALLY INTRODUCE AN EVEN BIGGER DIFFERENCE: DOMINANCE STATUS. IT IS WELL KNOWN THAT DOMINANCE INFLUENCES BEHAVIOURS, ESPECIALLY THOS STUDIED IN THIS STUDY, SO COMPARING ONLY SUBORDAINTE FOUNDRESSES AND DOMINANT SOCIAL PARASITES IS NOT CORRECT. WITHOUTH A COMPARISON WITH DOMINANT QUEENRIGHT QUEENS, YOUR STUDY SIMPLY DEMONSTRATE DIFFERENCE BETWEEN TWOP INDIVIDUALS WHICH ALSO HAVE A DIFFERENT DOMINANT STATUS.

REPLY 25/10/2021: I do not think this comment applies. The two females share a nest with its microclimate, and, for example, notwithstanding their dominance/subordinance status, the subdue foundress spent more time on the front side of the nest than  behind it, and the opposite was true for the parasite. This is thoroughly discussed in the manuscript.

---------------------------

In addition, the number and size of the larvae would differ between nests; even adding the number of larvae as a covariate will fail to control for larval hunger condition and for larval developmental phase.

THIS COUDL CLEARLY HAVE AN IMPACT, BUT IF THIS WAS A REASON TO NOT PERFORM COMPARISON MUCH OF THE RESEARCH ON SOCIAL INSECTS WOULD NOT HAVE BEEN DONE!

REPLY 25/10/2021: I am not saying that. Each model system has advantages and disadvantages. Here, I took advantage of a particular condition offered by parasitized social-wasp nests: the behavior of parasite and host is easily observable and they live on the same (very small) nest, exposed to the same microclimatic conditions.

---------

ADDING QUEENRIGHT CONTROL COLONIES (IN ADDITION TO SAME NEST FOUNDRESSES) WOULD HAVE ALLOWED TO INFER THE IMPORTANCE OF AMONG COLONIES DIFFERENCES.

REPLY 25/10/2021: In this work I was not interested in testing for among-colony differences. 

-----------

  • interpretation of the behavioural differences as adaptations for dehydration. You consider as indicative of adaptation against dehydration due to low CHC amount:
  • reduced activity
  • bias in position on the nest (behind rather than on frontal part)
  • less foraging
  • bias toward receiving than giving trophallaxis

While I see how each of this items could help in reducing water loss, this is not the unique interpretation. Actually, all of them are quite consistent with the "social parasite" syndrome/behaviour: dominant exploiters of a social host.

REPLY: I acknowledged that at lines 338-343.

IN MY OPINION ADDING A FEW SENTENCES IS NOT ENOUG IF THEN THE ENTIRE PAPER (TITLE ABSTRACT ETC) IS PRESENT THE PREFERRED HP AS THE MAIN AND MOST LIKELY INTERPRETATION OF THE RESULTS.

REPLY 25/10/2021: I made a hypothesis, I tested it and I reported the interpretation.

---------

For a social parasite it pays to avoid foraging (or spending less time foraging), resting, and receive sugary liquid food, even if this is not related to water loss. I am not saying it is not, simply that there are other possible explanations (energy related, dominance related, simply lifestyle related) which cannot be discarded. You argument a little bit this , at lines 325 and following, but in my opinion this is not sufficient to then orient the interpretation toward these behaviours being adaptations against water loss. Here, proper experiments and/or correlational data are needed to disentangle the several possible explanations. For example: a) young individuals do not have immediately a full chemical profile...hod do they behave?; b) are these behaviours influenced by dominance level of the social parasite? c) is the magnitude and direction of these behavioural differences influenced by climate (e..g humidity etc)?

REPLY: these interesting questions might be addresses in future studies.

THESE ARE ACTUALLY EXPERIMENTS THAT ARE NEEDED HERE, TO DISENTANGLE THE SEVERAL HP AND SUPPORT WHAT THE MS CLAIMS TO HAVE FOUND. I AM ACTUALLY SUGGESTING WAYS TO PROVE THE MS HP, NOT ADDITIONAL EXPERIMENT TO BE DONE IN THE FUTURE

REPLY 25/10/2021: I think the question I asked was properly tested here. I am also convinced that you are suggesting interesting questions that can be addressed in future studies.

---------

Also, I am wondering whether other social parasites with no reduced CHCs show nonetheless these behaviour. If possible to be done, this could be a revealing experiment.

REPLY: this interesting question might be addresses in future studies.

AS ABOVE, THIS IS SOMETHING I THINK IT IS NEEDED IF YOU WANT TO PRESENT AN INTERESTING HP (BUT ONE OF MANY) AS SOMETHING PROVED (YOU TALK ABOUT A TEST IN THE ABSTRACT)

REPLY 25/10/2021: The amount of research on the behavior of social parasites integrated into host colonies is rare. Research on the behavior of social parasites that are still chemical insignificant after having integrated host colonies is very very very rare. The model system I studied offered therefore a very rare opportunity. 

--------

My point here is that the correlational data you present might suggest strategies against water loss, but not only, and while you suggest something in the discussion the ms is too tailored on the water loss hypothesis. I do not think this is proven in your manuscript.

  • Aspects which have not been taken into account but might be really significant for the

It sems to me you have data along a somehow long period, from June to August. Is there any effect of timing? this might allow you to differentiate between a dominance hp and a water loss hp.

REPLY: Time of the day and period during the colony cycle were entered in the analyses.

YES, I SAW THAT, BUT THEY ARE NOT REALLY ANALYSED (TIME OF THE DAY IS, TIME FROM USURPATION OR TIME IN THE SEASON IS NOT) NOR I CAN SEE GRAPHS. YOU ARE BASICALLY CONTROLLING FOR THESE VARIABLES, WHILE WHAT I AM SUGGESTING IS TO USE THESE FACTORS TO TRY TO DISENTANGLE THE POSSIBLE HPs YOU HAVE (E.G. DOMINANCE AND WATER LOSS)

REPLY 25/10/2021: What do you mean by “they are not really analysed”? Isn’t a GZLMM a way to test the effects of predictors on a response variable? As reported in the manuscript “The predictors population, time of the day and period during the colony cycle, entered in the preliminary models, were removed from subsequent models when they were non-significant after comparing the competing models’ AIC values; results are reported from the models with the lowest AIC (Akaike Information Criterion) value to avoid information loss.”. I do not see the point of including graphs of variables that were removed from the analyses because non-significant.

---------

Both factor might change over time (temperature and humidity change, and also dominance and integration into the colony changes, form the very first days to the last days...) and not necessarily i the same direction. Thus, understanding IF and HOW these behaviours (foraging, trophallaxis, activity etc) change over time might allow you to better understand the related advantages.

REPLY: these questions might be addresses in future studies.

AS ABOVE, I THINK THESE ARE QUESTIONS THAT SHOULD BE ADDRESSED HERE.

REPLY 25/10/2021: The predictors time of the day and period during the colony cycle were included in the analysis to control for changes over time.

----------

  • General framework and wording

Overall, as explained above, I think the data are interesting but not unequivocally supporting your claims, as on the contrary your framework and wording suggest.

For example, you say the ms wants to test (lines 120, 162). I would say "investigate", or "explore", as you predictions are not so specific to allow a TEST of the hypotheses, as alternative explanations might be proposed and as the data are purely correlative. Also, at lines 359,361; 383-385 the unique interpretation given is that of the water loss balance, which to me is a speculation.

REPLY: I acknowledged alternative explanations at lines 338-343.

AS ABOVE, ONLY A FEW LINES IN MY OPINION DO NOT CHANGE THE OVERAL FRAMEWORK OF THE MS WHICH STILL PRESENT WATER LOSS AS THE MAIN HP AND ALMOST PROVEN FACT...IN MY OPINION IT IS NOT.

REPLY 25/10/2021: I do not think these analyses “proved” anything. I tested a hypothesis with statistical tests and acknowledged this is correlative evidence.

----------

  • Parasite rarity and the possibility to do

I see the concern here, and I appreciate the care about this aspect. However, I also think that if a model system does not allow a proper and sound test, not necessarily the test should be carried out. I mean, you coudl gather these important data, and simply explain them in light of several alternative hypotheses, suggesting your preferred one (and why) but leaving open the interpretation to the reader (maybe also suggesting what could be done if more parasite were available). in this way you would simply report and discuss the scientific evidence.

REPLY: Thanks for the suggestions.

IT SEEMS TO ME THAT THE SUGGESTION HAS NOT BEEN TAKEN INTO CONSIDERATION. I WOULD NOT HAVE ANY ISSUE IN ACCEPTING A MS PROPOSING JUST THE DATA AND THEN REPORT MANY HP AS THE POSSIBLE EXPLANATION, WHILE AS IT STANDS IT IS PRESENTED AS A TEST FOR THE HP OF WATER LOSS: IN MY OPINION THE EXP. DESIGN DOES NOT ALLOW THAT.

I am sorry to disappoint on this occasion with my non-positive evaluation. I really think the topic is nice, and surely there has been significant work behind. However, I do not think you present enough and unequivocal evidence for your claims.

AGAIN, I REALLY THINK THERE IS NICE AND INTERESTING MATERIAL IN THE STUDY, SIMPLY, I DO NOT THINK THE HP OF WATER-LOSS IS PROVEN BY YOUR EXP DESIGN AND DATA.

REPLY 25/10/2021: As above, I do not think these analyses “proved” anything. I tested a hypothesis with statistical tests and acknowledged that this was correlative evidence.

-----------
